# Rice-Associated Rhizobacteria as a Source of Secondary Metabolites against *Burkholderia glumae*

**DOI:** 10.3390/molecules25112567

**Published:** 2020-05-31

**Authors:** Giann Carlos Peñaloza Atuesta, Walter Murillo Arango, Jordi Eras, Diego Fernándo Oliveros, Jonh Jairo Méndez Arteaga

**Affiliations:** 1Chemistry Department, Faculty of Sciences, University of Tolima, Ibagué 730006299, Colombia; gcpenalozaa@ut.edu.co (G.C.P.A.); dfoliverosg@ut.edu.co (D.F.O.); jmendez@ut.edu.co (J.J.M.A.); 2Chemistry Department, Scientific Technical Services-TCEM, University of Lleida, 25198 Lleida, Spain; eras@quimica.udl.cat

**Keywords:** *Burkholderia glumae*, rice, metabolome, rhizobacteria, PGP

## Abstract

Various diseases, including bacterial panicle blight (BPB) and sheath rot, threaten rice production. It has been established that *Burkholderia glumae* (*B. glumae*) is the causative agent of the above mentioned pathologies. In the present study, antagonistic activity, growth promotion, and the metabolite profiles of two rhizobacteria, isolated in different paddy fields, were assessed against *B. glumae*. Strains were identified based on 16S rRNA gene sequences, and the phylogenetic analyses showed that both strains belong to the genus *Enterobacter*, with high similarity to the strain *Enterobacter tabaci* NR146667.2 (99%). The antagonistic activity was assessed with the disc diffusion method. Active fractions were isolated through a liquid/liquid extraction with ethyl acetate (EtOAc) from the fermentation media, and their antibacterial activities were evaluated following the Clinical and Laboratory Standards Institute (CLSI) guidelines. The Pikovskaya modified medium was used to test the ability of in vitro inorganic phosphorus solubilization, and BSB1 proved to be the best inorganic phosphorus solubilizer, with a solubilization index (SI) of 4.5 ± 0.2. The glass-column fractionation of the EtOAc extracted from BCB11 produced an active fraction (25.9 mg) that inhibited the growth of five *B. glumae* strains by 85–95%. Further, metabolomic analysis, based on GC–MS, showed 3-phenylpropanoic acid (3-PPA) to be the main compound both in this fraction (46.7%), and in the BSB1 extract (28.6%). This compound showed antibacterial activity against all five strains of *B. glumae* with a minimum inhibitory concentration (MIC) of 1000 mg/L towards all of them. The results showed that rice rhizosphere microorganisms are a source of compounds that inhibit *B. glumae* growth and are promising plant growth promoters (PGP).

## 1. Introduction

Rice provides a minimum of 20% of the daily calorie intake for more than half of the world’s population and is therefore of great importance in meeting the environmental challenge that food security has become [1,2,3]. Unfortunately, rice production around the world is threatened by several diseases, and bacterial panicle blight (BPB) is among the most problematic. *Burkholderia glumae (B. glumae)* has been recognized as the causal agent of this phytopathology. The most important symptoms of the disease include sheath rot, seedling blight, and grain rotting [4,5,6,7]. The occurrence and development of *B. glumae* are dependent on high nighttime temperatures, as well as on high humidity. Therefore, current climate change conditions favor its growth and the losses it causes in paddy fields [8,9].

Environmental conditions and the lack of effective control methods [10] are some of the reasons why BPB has become a severe disease. Oxolinic acid is so far the only effective chemical treatment against *B. glumae*; however, the emergence of resistant strains has limited the application of this treatment [11]. For these reasons, it is essential to find an effective method to control the pathogen. Modern agricultural production demands environmentally friendly agroecosystems that do not cause the simplification of soil biotic communities, and optimize crop productivity [12,13,14]; therefore, the traditional use of agrochemicals for the control of phytopathogens is now changing towards alternatives that promote low environmental impact agriculture [15]. In this regard, the complex rhizospheric ecosystem has become a source of microorganisms with a proven ability to control plant pathogens [16,17], as plants interact with the surrounding soil, by way of their roots and driven by exudation, to carry out the hundreds of essential physiological process that are mediated by microbial communities [18]. This plant-associated microbiome is responsible for plant development and is essential to the plant’s health [19].

The dynamic plant–microbiota relationships give rise to complex processes, which range from positive interactions, with beneficial microorganisms that promote plant growth or protect against pathogens, to negative interactions with pathogenic microorganisms [14,20]. The heterogeneous group of microorganisms that positively influences plant growth is called plant-growth-promoting rhizobacteria (PGPR) and can directly or indirectly improve plant growth [21]. PGPR can indirectly improve plant growth by inhibiting or preventing those detrimental effects caused by pathogens. This action on soil-borne pathogens may occur through different PGPR–pathogen interactions, for example, interference in biofilm formation, degradation of pathogenicity factors, or antagonistic activity, among others. Antagonistic activity is mediated either by small molecules with bactericidal activity or by the secretion of hydrolytic enzymes [22,23].

Many plant-growth-promoting bacteria (PGPB) with antagonistic activity against pathogenic bacteria and fungi have been isolated from plant surfaces, the soil, and the rhizosphere [10,21], as well as from the rice crop rhizosphere and leaves. The features of a rice crop make it a particular habitat for the growth of a wide variety of microorganisms [2,11]; for instance, two species of *Bacillus oryzicola* (YC7007-YC7010) were isolated from the rice root in paddy fields in Korea with antimicrobial activity. The YC7007 strain inhibited the growth of *Xanthomonas oryzae* pv. *oryzae* (KACC 10208), *B. glumae* (KACC 44022) [7], and *Fusarium fujikuroi*, the causal agent of Bakanae disease. On the other hand, *B. oryzicola* YC7007 showed the capability to induce systemic resistance against *F. fujikuroi* [24].

Three *Streptomyces* strains were isolated from the rice rhizosphere in paddy fields in Tolima (Colombia), these strains were tested against a wide number of bacterial and fungal phytopathogens and the A20 *Streptomyces* strain showed bactericidal activity against several *B. glumae* strains [25]. As mentioned earlier, rice leaves have also become a source of microorganisms with antagonistic activity; in Louisiana, bacteria identified as *Bacillus spp*. were isolated from the leaves of rice crops, these strains showed antifungal and antibacterial activity against *Rhizoctonia solani* and *B. glumae*, respectively [10].

Further, the production of secondary metabolites from rhizobacteria with antagonistic activity has been reported. Such is the case of the pyocyanin pigment produced by *Pseudomonas aeruginosa*. This compound showed antifungal activity toward *Fusarium oxysporum* Schlech [26]. The IC 1270 *Enterobacter agglomerans* strain has demonstrated its ability as an antagonistic microorganism toward fungal and bacterial phytopathogens. This strain produces a pyrrolnitrin-type antibiotic with fungistatic activity toward *Alternaria* sp., as well as significant bactericidal activity against *Agrobacterium tumefaciens* [27]. The *Streptomyces* species from the Enterobacteriaceae family produces the catecholate-type enterobactin, this compound was considered to be a characteristic metabolite of this family, and exhibited iron-chelating activity [28].

The objective of the present study was to identify the metabolites produced by rice-associated rhizobacteria, which could inhibit the growth of various strains of *B. glumae*. For this purpose, rhizobacteria from two different paddy fields were isolated and chosen according to their antagonistic capacity against *B. glumae*, and subsequently used to ferment a liquid medium. Further, the bioassays-guided method was followed to attempt to isolate bioactive fractions and their associated metabolites. Finally, the metabolomic analysis was carried out to characterize antimicrobial compounds, and we subsequently proved that the main compound, identified in both ethyl acetate (EtOAc) extracts, was one of those responsible for the bactericidal activity against *B. glumae*.

## 2. Results and Discussion

Twenty-six rice-associated rhizobacteria were isolated from the two assessed rice crops. From the lot with symptoms of *B. glumae*, thirteen strains were isolated and identified as BCB, with values ranging from 1 to 13. From the lot free of the symptoms caused by the pathogen, thirteen strains were isolated and identified as BSB, with values ranging from 1 to 13.

### 2.1. In Vitro Antagonistic Activity of Isolated Rhizobacteria

The 26 rice-associated rhizobacteria were screened for their capacity to inhibit the growth of five *B. glumae* strains, in a dual culture in vitro assay. However, only two isolates, identified as BCB11 and BSB1, displayed antagonistic activity on at least two pathogen strains (Table 1). BCB11 showed the highest levels of activity, with an inhibition of the pathogenic growth zone ranging from 7.6 mm, toward strain 3252-8, to 9.1 mm, for strain 296.

BSB1 also expressed antagonistic activity, with inhibition zones of 6.7 mm, toward strain 296, 5.7 mm toward strain 448, and 6.8 mm toward strain 3252-8. In dual confrontations, these strains were found to exert moderate antagonistic activity, as compared to oxolinic acid (50 mg/L). When the halozones produced by the rhizobacteria were relativized with those produced by the oxolinic acid, we found that the clear zones caused by BCB11 towards strains 296 and 3252-8 of *B. glumae* reached 41.2% and 36.2%, respectively. BSB1 showed the ability to inhibit the growth of strains 296, 448, and 3252-8 with halozones that represented 30.3%, 32.9%, and 34.4% of the inhibition caused by the oxolinic acid, respectively.

### 2.2. Activities that Promote Plant Growth and Promising Strain Identification

#### 2.2.1. In Vitro Rice Seed Germination and Seedling Growth

The effect of the rice-associated rhizobacteria on rice germination and seedling growth was evaluated in vitro. Seeds from “FEDEARROZ 2000” (F-2000) (susceptible to *B. glumae*) and “FEDEARROZ 67” (F-67) varieties were inoculated with BCB11 and BSB1 antagonist strains. The greatest effect was observed in the germination rate in F-67 seeds. BSB1 was able to promote an enhanced germination rate, which reached 27.7%, in comparison to noninoculated seeds. Conversely, seeds inoculated with a BCB11 suspension did not show significant differences in germination rates, as this parameter was the same with and without the presence of this bacterial inoculum (Table 2).

On the other hand, bacterial suspensions of BSB1 and BCB11 did not affect biomass production in any of the rice varieties, given that seedling dry weights were not significantly different from those in the untreated control. Statistically significant differences in dry weight were found between rhizobacteria treatments in the F-67 variety. Seeds inoculated with the BSB1 strain increased their biomass production by 12.4% compared to the BCB11 inoculated treatment, while the F-2000 variety did not display biomass differences based on the bacterial treatments employed.

The BSB1 bacterial suspension affected seedling development in F-67, which resulted in a 26.49% decrease in shoot length, as compared to the control. Similarly, the BCB11 strain caused a 15.21% decrease in root length in F-2000. The shoot/root ratio showed that F-2000 had greater root development than seedling vegetative development, marked by a shoot/root relationship of less than one. Conversely, F-67 showed greater vegetative development compared to root development. This is likely the result of genetic differences between the varieties evaluated.

It is important to highlight that the plant growth promotion experiments were performed without a substrate; both treatments and untreated controls were only supplied with a 10 mM solution of MgSO_4_, and after germination, no nutrients were used to improve seedling growth. This fact did not allow determining the capacity of rhizobacteria to transform, mobilize, or solubilize nutrients to provide the plant with essential resources for its development. However, despite these characteristics of the experiment, the behavior of the two strains under in vitro conditions showed no phytotoxic activity on seedling development, although their inoculation did not result in a clear improvement in the growth factors of rice varieties F-67 and F-2000.

#### 2.2.2. In Vitro Inorganic Phosphate Solubilization

The ability to solubilize inorganic phosphorus was assessed in vitro in Pikovskaya agar augmented with Ca_3_(PO_4_)_2_. Phosphorus solubilization is just one trait of bacteria that may promote plant growth [29]. Bacterial isolates were able to solubilize inorganic phosphorus, and this capacity was detected by the production of a clear zone surrounding the BCB11 and BSB1 colonies (Figure 1).

The phosphate solubilization index (SI) was calculated and determined to be 4.5 ± 0.3, with BSB1, and 2.6 ± 0.2, with BCB11. Phosphate solubilization screening in a plate assay showed BSB1 to be more effective in its formation of a visible halozone on agar plates and demonstrated capability approximately two-fold higher than BCB11. With the GC–MS analysis, the presence of 3-indoleacetic acid (IAA) in the EtOAc extracts of the ferments produced with both rhizobacteria was identified, demonstrating that the relative content of IAA was four-fold higher in the ferment produced with BSB1 than BCB11. The *Enterobacter* isolates of this study appear promising as reflected by the phosphate solubilization potential and the presence of IAA in the EtOAc extracts.

Phosphorus solubilization is a direct mechanism used by PGPR bacteria to improve plant growth [30]. Rhizosphere bacteria, which have a demonstrated capability for inorganic phosphorus solubilization and indole acetic acid (IAA) production, promote plant growth in beans and maize. Bacterial strains isolated from rhizosphere soil from beans, maize, and rice were tested for inorganic phosphate solubilization and IAA production. Here, the *Klebsiella* SN 1.1 strain was the best IAA producer, and it solubilized a great deal of phosphate in an optimized culture and demonstrated the ability to enhance bean growth in vivo [31].

The generally accepted mechanism by which phosphorus-solubilizing bacteria exert this action is associated with the production of low-molecular-weight organic acids [32], in this regard, a high content of organic acids was found in the EtOAc extracts in both fermentations. The organic acid content in the BSB1 extract was 35.1%, despite being a total extract, and, in the (F4) subfraction of BCB11, it was 52.3%. This fact could explain the high SI obtained with the BSB1 strain. The genus *Enterobacter* has also been recognized to promote plant growth and one of its features is the solubilization of inorganic phosphorus [23,32].

#### 2.2.3. Identification of Promising Strains

The rhizobacteria isolated from the rice rhizosphere were identified by partial 16S rRNA gene sequence analysis. The 16S rRNA gene sequence of the two strains was a continuous segment of 1107 base pairs, and two alignment regions, totaling 100 bp, were excluded from the beginning and end of each data set in order to eliminate gaps due to missing data. The gene sequences of the two strains were aligned identically except for a single nucleotide (Appendix A) and based on the comparative analysis with the data available in GenBank. Using the BLAST (Basic Local Alignment Search Tool) homology search, the two strains showed the greatest similarity to *Enterobacter tabaci* NR146667.2 and *Enterobacter asburiae* NR024640.1 with identity percentages of 99.12% and 99.03% for BSB1 and 98.92% and 98.65% for BCB11, respectively. The BSB1 and BCB11 strains are in the same clade and the branch length of the phylogenetic tree is the same for both strains (Figure 2 and Appendix A).

Although the phylogenetic analysis showed that the two isolated strains in this work presented similarity with *Enterobacter tabaci*, the evaluation of the metabolomic profile showed some differences between them (Table 3). This result can have one of two causes: the first is that *B. glumae* triggered some metabolic differentiation between the isolated rhizobacteria; and the second is that the result would be associated with two different strains of *Enterobacter*. Resolving this ambiguity requires the use of more specific tools such as multilocus sequence analysis (MLSA) [33]. The active strains isolated in this work belong to the Gram-negative group. This type of bacteria is commonly found in the soil, and has been reported to promote plant growth [21,34].

### 2.3. EtOAc Extract Antibacterial Activity and Metabolomics Profiles

#### 2.3.1. Antibacterial Activity toward *Burkholderia glumae*

The bioactive metabolites produced during BCB11 and BSB1 fermentation were isolated from the fermented biomass using liquid/liquid extraction with EtOAc. Following solvent evaporation, 1.16 g and 0.32 g of extract were obtained. The EtOAc extracts were emulsified to guarantee their solubilization in water (Appendix A); 2500 mg/L solutions were prepared, and subsequently evaluated against five *B. glumae* strains.

At 2500 mg/L, the total EtOAc extract from BSB1 fermentation, as well as the active subfraction (F4) from BCB11 fermentation, inhibited the growth of all *B. glumae* strains above 90% (Figure 3). However, against the 453 strain, the purified subfraction (F4) of the ferment made with BCB11 only reached 85.6 ± 0.6%. The liquid/liquid extraction with EtOAc [35] isolated metabolites that are likely to be the reason for the inhibitory activity in dual confrontations.

The use of the microdilution method to evaluate inhibitory activity established that, unlike that which occurred with dual confrontations, EtOAc extracts at 2500 mg/L showed inhibitory activity against all pathogenic strains of *B. glumae*. The concentration of the metabolites in the extracts may be the reason for this, or possibly a diffusion problem of the metabolites in the agar.

#### 2.3.2. Metabolomic Profiles in Rice-Associated Rhizobacteria with Antagonistic Activity

Two *Enterobacter* strains from the rice rhizosphere, isolated from two different paddy fields, were studied to determine their metabolomics profiles. A nontargeted profiling method was used to identify as many metabolites as possible in the EtOAc extracts, which resulted from fermentations performed with these rhizobacteria.

The EtOAc extract from BCB11 fermentation was dark brown in color, and the EtOAc extract from BSB1 fermentation was yellow. The bioassay-guided fractionation of the BCB11 EtOAc extract produced four fractions that were identified as 1, 2, 3, and 4, and they were assessed against five *B. glumae* strains. The antibacterial active fraction was 1; this fraction was subjected to further isolations with EtOAc/Hexane (1:1), eight subfractions were obtained (F1–F8), and the antimicrobial compounds were found in subfraction F4. The solvent’s polarity suggested that the active metabolites were hydrophobic. GC–MS was used to analyze the F4 subfraction (Table 3), through which 28 compounds were identified using the NIST17.L, NIST MS Search 2.3, and WYLEY275 databases.

#### Metabolomics Profiles in BCB11 EtOAc Extract

All compounds found in the F4 subfraction (Table 3) were identified as trimethylsilyl derivatives. Subfraction F4 showed interesting inhibitory activity against all *B. glumae* strains (Figure 3a), which demonstrates the presence of bioactive compounds. The remaining fractions (2, 3, and 4) and subfractions (F1–F3 and F5–F8) were not analyzed by GC–MS, as they did not display inhibitory activity toward the pathogen. Among the compounds identified, 33% were aromatic, and their relative areas were equivalent to 61.8% of the total detected peaks. Two phenyl alkanoic acids were identified and assigned to Compounds **3** and **10**. Compound **3** was identified as benzeneacetic acid. Relevant antibacterial activities have been reported for this compound toward *Escherichia coli* and *Ralstonia solanacearum* [36].

The relative area of Compound **10** was 46.7% of the detected substances and was the main constituent of the F4 subfraction from the BCB11 EtOAc extract. This compound was known as 3-phenylpropanoic acid (3-PPA or hydrocinnamic acid), with molecular ion *m*/*z* 222.1 (27.3%) (Appendix A). 3-PPA demonstrated its ability to inhibit the normal development of various test microorganisms. With a minimum inhibitory concentration (MIC) of 10 µg/mL, it inhibited the development of *Pseudomonas aeruginosa* and *Pseudomonas fluorescens*. At 100 µg/mL, it also inhibited *Klebsiella pneumoniae* and *Staphylococcus aureus*, among others [37].

The identity of the main compound in the F4 subfraction was verified using FT-IR and NMR. The carboxylic acid carbonyl was displayed in the infrared spectrum of this purified subfraction, at 1741.4 cm^−1^. The absorption bands between 3015.7 and 3020 cm^−1^, and between 650 and 850 cm^−1^ were suggestive of aromaticity (Appendix A), which was further confirmed by the presence of the corresponding signal in the nuclear magnetic resonance spectra. The ^1^H-NMR spectrum showed signals at δ 7.21 and δ 7.29 that could be associated with aromatic protons; additionally, the signals at δ 2.68 and δ 2.96 may be due to methylenes α and β to the carboxyl group (Appendix A). Compound **17** was accepted as cinnamic acid. The potential antimicrobial and antifungal activities of cinnamic acid and several of its derivatives are well known [38,39]. Compounds fifteen and nineteen were known as phenyl amides and their relative areas totaled 7.42% of the detected compounds. Compound **15** was identified as *N*-(2-phenylethyl)acetamide and Compound **19** was identified as *N*-phenethylpropionamide.

The main phenyl amide found in the EtOAc extract was *N*-(2-phenylethyl)acetamide, with 7.2% of the total area of the phenyl amide compounds. The IR spectrum showed a band at 3463.7 cm^−1^, which may be assigned to secondary amides. Extracts of EtOAc obtained from *Stropharia semiglobata*, which contained *N*-(2-phenylethyl)acetamide, 1*H*-indole-6 methyl, 3-methyl-2(1*H*)-quinolinone, and ethyl(3,5-dihydroxyphenyl)acetate, showed moderate activity toward *Bacillus subtilis*, *Pseudomonas aeruginosa*, *Escherichia coli*, *Escherichia faecalis*, and *Staphylococcus aureus* [40].

Peaks 18, 20, and 21 were assigned to other aromatic compounds. Peak 18 was accepted as tyrosol; this compound constituted 0.95% of the compounds found in the sample. The IR spectrum showed bands at 1047, 1097, and 847.3 cm^−1^, which indicated the presence of disubstituted aromatic compounds (Appendix A). Tyrosol is crucial to cell division, and constitutes an important density-sensing molecule for the growth and development of *Candida albicans* [41], the bactericidal activity of olive oil aqueous extracts, containing tyrosol, against *Helicobacter pylori* was reported [42]. The use of essential oils, with tyrosol content, also showed antibacterial activity toward *Listeria monocytogenes* [43].

Peaks 20 and 21 were recognized as phenolic compounds and their relative area was 3.3%. Peak 20 was admitted as 4-hydroxybenzeneacetic acid. This compound has been isolated from *Lucilia sericata* maggot extract, and the bactericidal activity was demonstrated via zone inhibition assay, toward *Micrococcus luteus* and *Pseudomonas aeruginosa* [44]. Compound **21** was known as phloretic acid (Appendix A). It was shown that concentrations of 1000 µg/mL of the compounds 4-hydroxybenzeneacetic acid and phloretic acid have inhibitory effects on CECT 5947 and lpxC/tolC *Escherichia coli* strains, *Lactobacillus paraplantarum*, *Lactobacillus plantarum*, and *Lactobacillus coryniformis* among other microorganisms [45].

Several other compounds for which bactericidal bioactivity has been reported in the literature were found. Compound **1** was accepted as lactic acid, the mediation of lactic acid in the bactericidal activity exerted by *Lactobacillus rhamnosus* GG toward *Salmonella typhimurium* was reported [46,47]. Compound **24** was identified as tryptophol, this compound is an indole-based metabolite produced by plants, bacteria, fungi, and sponges. Its origin is closely related to the metabolic pathways of tryptophan and tryptamine, and it has shown biological activity toward various living organisms, including *Campylobacter jejuni*, a Gram-negative bacillus that causes bacterial gastroenteritis in humans [48]. Tryptophol production has been reported in the phytopathogen fungus *Ceratocystis adipose*, which also yields antibiotic [49].

Substances with fungicidal activity were also known. Compound **28** was admitted as (*Z*)-octadec-9-enenitrile (oleonitrile). Oleonitrile has been found in ethanolic extracts of the *Tetraselmis tetrathele* microalgae, which is grown under varying salinity conditions [50]. In terms of its biological activity, ether extracts of endophytic fungus from *Panas ginseng*, with 1.38% of oleonitrile, displayed antifungal activity toward *Candida albicans*, *Cryptococcus neoformans*, *Trichophyton rubrum*, and *Aspergillus fumigatus* [51].

Compounds **26** and **29** were recognized as auxins. The relative areas of these compounds sum to 0.45% of the total. Peak 26 was identified as 3-indolacetic acid (IAA) with molecular ions at *m*/*z* 319 (24.4%). On the other hand, compound **29** was known as 3-indolepropionic acid (IPA) with molecular ions at *m*/*z* 333.1 (31.3%). The auxins are considered phytohormones that improve plant growth because they stimulate cell division and differentiation [31,52,53].

#### Metabolomics Profile in BSB1 EtOAc Extract

One hundred and thirty-three peaks were detected in the total EtOAc extract of *Enterobacter sp*. Twenty-eight compounds were identified (Table 3 and Appendix A), and the remaining compounds were detected, but not identified. The metabolomic analysis of the substrate used for BSB1 fermentation established that 33% of the identified compounds were aromatics, and their relative areas were equivalent to 38.9% of the total compounds found after the fermentative process. Compound **5** was identified as catechol, in addition to the significant correlation found between the catechol concentration and antioxidant activity [54], its antibacterial activity toward the phytopathogen bacteria *Xylella fastidiosa* was also demonstrated [55].

Compound **14** was accepted as the 4-hydroxybenzyl alcohol; this compound was also identified in the EtOAc extract of the fermented biomass with BCB11. This alcohol is an effective neuroprotective agent against different cerebral and nervous disorders [56,57,58]. In the same way as the EtOAc extract from BCB11, the main compound found in the metabolomic profile of BSB1 was 3-PPA (Compound **10**), and the relative area corresponded to 28.6% of the detected compounds (Appendix A).

Several compounds with indolic nuclei were found in the metabolomic profile of BSB1, and their relative areas encompassed 5.9% of all compounds. Compound **12** was identified as indole with a relative area of 3%. Compounds **24** and **26** were previously known in the EtOAc extract from BCB11 as tryptophol and IAA, respectively. Compound **30** was accepted as 5-hydroxytryptophol, with a relative area of 0.72%. A great variety of biological activities have been attributed to compounds that possess indolic nuclei, from antifungals and antimicrobials to antihypertensives and anticancer drugs; this compound was also found in the fermented biomass of BCB11 [59]. Approximately 32% of the metabolites identified in the EtOAc extracts from the ferments of the *Enterobacter* strains isolated in this work, which grew in TSB medium, were compounds with a reported bacterial activity that supports the inhibitory activity of these strains against *B. glumae*.

#### Antibacterial Activity of the Main Compound Found in EtOAC Extracts, 3-Phenylpropanoic Acid, against *B. glumae*

As mentioned above, 3-phenylpropanoic acid was found to be the main compound in both EtOAc extracts; this fact led us to suppose that this metabolite was responsible for the inhibitory activity exerted by EtOAc extracts against the pathogen, and therefore its antibacterial potential was tested against all five strains of *B. glumae*. The MIC values are shown in Figure 4. A comparison of the sensitivities of the *B. glumae* strains to 3-phenylpropanoic acid indicated that there was no difference in the inhibitory effect as the MIC values were 1000 mg/L in all cases.

It is important to remark that 3-phenylpropanoic acid showed inhibitory activity at low concentrations only against 453 *B. glumae* strain (EF193638.1); however, it did not inhibit the growth of the phytopathogen completely. The evaluation of the antibacterial potential demonstrated that 3-phenylpropanoic acid is one of the metabolites responsible for the inhibitory effects exerted by the EtOAc extracts from fermented biomass with the rhizobacteria BCB11 and BSB1 towards *B. glumae*. This result is supported by the works of several authors who have reported the bactericidal activity exerted by phenolic acids against pathogenic bacteria [37,45,60,61,62].

Most of the identified compounds were aromatic, including the major compound, or had an indolic nucleus. This suggests that their production is carried out via shikimate, which not only results in the biosynthesis of the aromatic amino acids L-tryptophan, L-tyrosine, and L-phenylalanine, but also produces several aromatic precursors. This metabolic pathway is exclusive to bacteria, fungi, and plants [63,64]. Moreover, tryptic soy broth (TSB) contains digests of soybean meal and casein, making it a nutritious medium that provides aromatic amino acids and other nitrogenous substances [65]. It is therefore valid to assume that the isolated rhizobacteria have a metabolic capability for generating products with aromatic and indolic nuclei from this culture medium [66,67].

The metabolomic profiling of BCB11 and BSB1 allowed the identification of phenylacetic acid (benzeneacetic acid). Although their concentrations cannot be compared, it is possible to assume the presence of this compound due to the induction of indolepyruvate decarboxylase in the presence of tryptophan or phenylalanine in the fermentation medium. Likewise, the tyrosol found in EtOAc extracts suggests that indolepyruvate decarboxylase acted on the tyrosine; this amino acid can be synthesized from phenylalanine [68].

The L-tryptophan amino acid is a precursor for the biosynthesis of indole acetic acid (IAA) [31]. IAA is a phytohormone involved in different developmental process and can stimulate plant growth or improved tolerance to abiotic stress. It can also be a virulence factor produced by phytopathogens, all depending on their concentration and plant tolerance [68]. The IAA was found in EtOAc extracts from both strains, but the relative content in the extract from BSB1 fermentation (1.29%) was four times higher than in the extract from BCB11 fermentation (0.30%). The production of this metabolite has been reported by two mechanisms: the first one related to phytopathogenic bacteria via indole acetamide, and on the other hand, via indolepyruvic acid. The production of IAA through indolepyruvate decarboxylase is involved in plant growth promotion and this pathway was reported for *Enterobacter cloacae* and *Pseudomonas putida*—bacteria that inhabit the rhizosphere [15,53].

For the above reasons, it is possible to assume that the higher IAA content found with the BSB1 strain was responsible for the better response of growth factors and shoot and root length in the F-2000 variety, compared to the response of the seedlings to the inoculum with BCB11. Nevertheless, on the other hand, it could be responsible for the decrease of the length of the shoot in the variety F-67. Even though the metabolomic profile reported here is not complete, as only EtOAc was used for the extraction of the fermented biomass, and it was impossible to identify some compounds with important relative signals; this analysis allowed us to identify the main compound in the EtOAc extracts, 3-phenylpropanoic acid, which demonstrated significant bactericidal activity against five different strains of *B. glumae*.

## 3. Materials and Methods

### 3.1. Materials

To test the effect of the main compound putatively involved in antibacterial activity against *B. glumae*, antibacterial bioassay of the 3-phenylpropanoic acid was conducted as described in Section 3.7. The 3-phenylpropanoic acid (>98%) was purchased from Tokyo Chemical Industry Co. Ltd. (Zwijndrecht, Belgium). All other chemicals were of analytical grade.

### 3.2. B. glumae Strains

Pathogenic strains were supplied as follows: the 3200-12 (CP009435.1) and 3252-8 (TXID1248414) strains were provided by the phytopathology laboratories of International Centre of Tropical Agriculture (CIAT), Palmira-Valle del Cauca, Colombia and the 448 (EF193638.1), 296 (EF193638.1), and 453 (EF193638.1) strains were provided by the Federación Nacional de Arroceros (FEDEARROZ), Saldaña-Tolima, Colombia.

### 3.3. Sample Collection and Rhizospheric Bacteria Isolation in Rice

Twenty samples of rhizospheric soil were collected from the “Maja 6” crop variety from two different paddy fields, located in the municipality of Prado, Tolima, Colombia, on the “Tres claveles” farm situated at an altitude of 303 m above sea level (m a.s.l.) with an average temperature of 32 °C. The first paddy field (3°47′11.6”N-74°56′27”W) exhibited symptoms of *B. glumae* infection, and the second paddy field (3°45′42.3”N-74°56′37.9”W) did not. Ten rice plants were collected randomly from each field, placed in polyethylene bags, and immediately transported to the laboratory. Samples were air-dried at room temperature for two hours, and the nonrhizospheric soil was then manually separated by shaking. The remaining soil attached to the roots was considered as rhizospheric soil and was detached manually from the roots in a sterile vessel.

We weighed 1.0 g of rhizospheric soil and placed it in 9.0 mL of sterilized buffered peptone water (Merck^®^, Darmstadt, Germany). Aliquots were 10-fold serially diluted in autoclaved peptone–water, and the diluted samples were incubated at 30 °C. Following 48 h of incubation, 100 µL of each dilution step were spread on trypticase soy agar plates (TSA, Merck^®^, Darmstadt, Germany), single colonies were isolated and further purified by streaking on fresh TSA. Each bacterial strain was macroscopically differentiated, based on colony morphology. Purified bacterial strains were stored in glycerol (30%, *w*/*w*%) at −80 °C for further use.

### 3.4. Identification of Promising Strains

Two strains were selected based on their antimicrobial activity. RNA extraction was performed using standard methods, which involves the breaking down of bacteria by heat in the presence of a nonionic detergent [69]. The identification of strains with promising antagonistic activity was carried out by 16S rRNA sequence gene homology. The 1465 bp region of the 16S rRNA was amplified by PCR using 27F (5′AGAGTTTGATCMTGGCTCAG3′) and 1492R (5′TACGGYTACCTTGTTACGACTT3′) primers, the cleaned PCR products were sequenced using 337F (GACTCCTACGGGAGGCWGCAG), 518F (CCAGCAGCCGCGGTAATACG), 907R (CCGTCAATTCMTTTRAGTTT), and 1100R (GGGTTGCGCTCGTTG) primers, with specific PCR protocol. Macrogen Inc. (Seoul, Korea) sequenced the PCR products. The consensus sequences of 16S rRNA genes were generated from forward and reverse-sequence data, using the STADEN PACKAGE v.2.0 aligner software (Whitehead Institute for Biomedical Research, Cambridge, MA, USA). Sequences were aligned and subjected to a BLAST (Basic Local Alignment Search Tool, NCBI, Rockville, MD, USA) for nucleotide similarity search using the Genomic-NCBI databases, and latter submitted to the NCBI Genbank (NCBI, Rockville, MD, USA). The PCR reaction program was initiated at 94 °C for four minutes, followed by 30 cycles at 94 °C for 30 s, followed by 35 cycles at 55 °C for 35 s, 72 °C for 90 min, and a final extension at 72 °C for 10 min. Agarose gel electrophoresis for the detection of amplified RNA bands was carried out.

16S ribosomal RNA sequences were aligned with the ClustalW program, in MEGA X [70], and later used to construct a phylogenetic tree, with the neighbor-joining method and MEGA X software (Pennsylvania State University, State College, PA, USA). The phylogenetic relationship was constructed using the nearest-neighbor data analysis method, with 1000 bootstrap replicates.

### 3.5. Antagonistic In Vitro Activity Toward B. glumae Strains

Rhizobacteria antagonistic activity was assessed, and the diameter of the halo around the paper disc occasioned by the growth inhibition of *Burkholderia glumae* was measured as a positive response under the protocols of the Clinical and Laboratory Standards Institute [71]. The inhibition test was assessed using the paper disc-agar diffusion method, and Muller–Hinton Agar (MHA. Oxoid, Oxoid LTD, Basingstoke, England) was the culture medium. Before the determination, each rhizospheric bacteria and all strains of *B. glumae* were cultured in Luria–Bertani broth (LB) and kept in an incubator at 28 °C for 24 h, and the OD_610_ was determined. The bacterial absorbance of the cultures was then adjusted to OD_610_ nm = 0.2 with LB, in order to ensure a density equivalent to 10^7^–10^8^ cfu/mL. The test was performed by massively growing 100 µL of each *B. glumae* strain, on individual MHA plates, with L-shaped glass rods. Four paper discs (5.0 mm in diameter), which had been previously sterilized, were placed in each quadrant on the inoculated plates and then impregnated with 10 µL of the bacterial suspension to be tested. Measurements were taken in four different directions over the inhibition halozone, and reported as the average thereof.

### 3.6. Fermentation and Metabolite Extraction

Purified bacterial strains were fermented to obtain bioactive metabolites. The fermentation was performed in tryptone soy broth (TSB, OXOID, CM0129) following the preparation of a preinoculum. To this end, 10 µL of a bacterial suspension (OD_610 nm_ = 0.2) previously grown in LB, was added to 25 mL of TSB in a 100 mL Erlenmeyer flask, and was kept on a shaker at 30 °C for 24 h. The preinoculum was added to 175 mL of TSB in a 500 mL Erlenmeyer flask, the mixture was incubated for 48 h with shaking at 150 rpm (CERTOMAT^®^ MOD II, Sartorius, Goettingen, Germany) at room temperature. It was then centrifuged at 11,515× *g* (HERMLE Z326K, Labortechnik, Wehingen, Germany) to separate the biomass, and the fermented medium was subsequently extracted with EtOAc (Sigma-Aldrich, St. Louis, MO, USA) (3 × 100 mL) in a separatory funnel. Organic phases were combined and dried with Na_2_SO_4_. The EtOAc extract was evaporated to dryness under reduced pressure.

### 3.7. Antibacterial EtOAc Extract Activity toward B. glumae

Antibacterial activity was evaluated by measuring the inhibition of the growth of five strains of *B. glumae* caused by EtOAc extracts from the fermented biomass with rhizobacteria. The activity was evaluated through optical density (OD) measurements at 610 nm on 96-well plates, using a MULTISKAN^TM^ GO microplate spectrophotometer (THERMO Scientific^TM^, Vantaa, Finland). The extract solubilities and the spectrophotometric characteristics of the analysis imposed conditions on the test implementation. For these reasons, an extract with a sufficiently high concentration was tested, thus the existence of negative results would not be the effect of a low concentration and the resulting mixture of the extract with the culture medium (LB) would not have a high initial absorbance that would prevent the observation of small changes caused by the growth of the pathogen.

Each extract was assessed at a concentration of 2500 mg/L and eight replicates were performed. The EtOAc extracts were emulsified with TWEEN-80 (1% *v*/*v*), oily phase (sunflower oil) (25% *v*/*v*), and water (74% *v*/*v*) to ensure their solubility in the test medium. For this, the surfactant proportion was dissolved in water at 40 °C, the EtOAc extract weight was dissolved in the oily phase also at 40 °C, and an O/W emulsion was prepared by slowly adding the oily phase into the watery phase. Subsequently, a three-way valve was used to homogenize the dispersion. A blank was prepared using water instead of the EtOAc extract and was emulsified as described above. Extract stock solutions were prepared at 10,000 mg/L. Before the activity test, all of the *B. glumae* strains were grown in LB and kept in an incubator at 28 °C for 48 h. The suspensions were diluted with LB to ensure densities equivalent to 10^7^–10^8^ cfu/mL (OD_610 nm_ = 0.2). The test was performed by mixing 50 µL of each extract solution, 50 µL of the *B. glumae* pathogen suspension, and 100 µL of the LB medium into each well (×8 replicates), and the same was done for each pathogenic strain. A positive oxolinic acid control, at 50 mg/L, a growth control (50 µL pathogen strain + 50 µL emulsion without extract + 100 µL LB) and a contamination control (50 µL of autoclaved H_2_O + 50 µL emulsion without extract + 100 µL LB) were made on the same plate.

The antibacterial activity measurement was based on the pathogen growth kinetics, and the growth control curve slopes were compared to the treatment growth curve slopes. For this purpose, the absorbance was measured at time zero, immediately after inoculating the microplate with the pathogen. Next, the microplate was incubated at 28 °C, and optical density measurements were performed at 6, 12, and 24 h, with prior shaking. The antibacterial activity was calculated with the Equation (1)
Inhibition(%) = ((slope control curve − slope treatment curve)/slope control curve) × 100(1)

### 3.8. In Vitro Plant-Growth-Promoting Activity

#### 3.8.1. Seed Germination and Seedling Growth

We assessed the effects of rhizobacteria strain suspensions on the rice seeds germination rate and seedlings development of the F-2000 and F-67 varieties. The seed germination rates and growth parameters, such as biomass production, shoot length, and root length were measured in vitro in the same experiment.

The seeds were surface-sterilized via stirring in hypochlorite solution (1.2% *v*/*v*) for two minutes and were then washed three times in sterile distilled water. Subsequently, the seeds were placed in an ethanol solution (70% *v*/*v*) for two minutes, and finally, they were washed three times with sterile distilled water. To inoculate the seeds, rhizobacteria were cultured in an LB medium and kept in an incubator, at 28 °C, for 48 h. They were then centrifuged at 5600× *g* for 15 min. The supernatant was discarded and the pellet resuspended in a sterile 10 mM solution of MgSO_4_. Washing was performed twice and the absorbance was adjusted (OD_610 nm_ = 0.2) to create 10 mL of each bacterial suspension. The sterilized seeds were placed in each suspension separately, and stirred for 24 h.

The experiments were performed in sterile glass flasks (11 cm × 4 cm). To ensure the presence of each rhizobacteria during germination and seedling development, a previously autoclaved filter paper disc (4 cm in diameter) was put in the bottom of the flask, and then 2.0 mL of each bacterial suspension in MgSO_4_ was deposited on a paper disc, together with ten previously inoculated seeds. The controls were prepared by depositing 2.0 mL of a 10 mM MgSO_4_ sterile solution on the filter paper. Finally, the flasks were closed with Parafilm^®^ to prevent evaporation. These were then germinated for three days at 30 °C in the dark. After this period, the germination rate was measured.

Subsequently, the flasks were taken to a greenhouse for 15 days, with a 12-h light–dark photoperiod, and the flasks were then removed to measure the shoot and root length, as well as the seedlings dry weight. The dry weight was determined by drying the seedlings in an oven, at 60 °C, to a constant weight. All tests were performed in triplicate.

#### 3.8.2. Phosphate Solubilization Assessment

The ability of rhizobacteria to solubilize inorganic phosphorus is an important feature in the characterization of these strains as plant-growth-promoting microorganisms. The Pikovskaya medium (ammonium sulfate 0.5 g/L, potassium chloride 0.2 g/L, magnesium sulfate 0.3 g/L, iron sulfate heptahydrate 0.004 g/L, sodium chloride 0.2 g/L, glucose 10 g/L, yeast extract 0.5 g/L, tricalcium phosphate 5.0 g/L, agar 15 g/L, and bromocresol purple 0.1 g/L, pH 7.0) was used to test the ability of promising rhizobacteria to solubilize tricalcium phosphate (Ca_3_(PO_4_)_2_).

The test was performed using the paper disc agar diffusion method. Prior to determination, the rhizobacteria strains were grown in LB and kept in an incubator at 37 °C for 48 h. Cultures were diluted to ensure an equivalent density to 10^7^–10^8^ cfu/mL (OD_610 nm_ = 0.2). Four paper discs that were autoclaved (5.0 mm in diameter) were placed in each quadrant of the plates, and then inoculated with 5 µL of the bacterial suspension of each strain and kept in an incubator, at 37 °C for 24 h. Measurements were taken in four different directions of the halozone, and reported as the average of these. To determine the inorganic phosphate solubilization capacity, the solubilization index (SI) was calculated using Equation (2) [30].
SI = ((colony diameter + halozone)/colony diameter)(2)

### 3.9. Metabolites Extraction and Metabolomic Analysis

#### 3.9.1. Active Fractions Chromatographic Separations

The total EtOAc (1.16 g) extract obtained from the BCB11 fermentation medium was subjected to separation in a packed glass column (1.5 × 8.0 cm) with silica gel (MERCK^®^. Kieselgel 60G, Art. 7731), eluted with a step gradient EtOAc/MeOH (90:10, 80:20, 70:30, 60:40, 50:50, 40:60, 30:70, 20:80, 10:90, 0:100, each 80 mL) to yield four fractions (1–4). The fractions were collected based on their TLC profiles. The bactericidal activity toward the pathogen was then assessed, using the microplate dilution method.

Fraction 1 (25.9 mg) (90:10–70:30 EtOAc/MeOH) showed bactericidal activity and was then subjected to further isolations. This fraction was dried under reduced pressure, at 37 °C, and examined by TLC, with phosphomolybdic acid–ethanol, a major spot at *R*_f_ 0.8 was displayed (Si gel (MERCK^®^ 5554), EtOAc/Hexane (1:1)). Fraction 1 was loaded onto a silica gel column and was eluted using EtOAc/Hexane to give eight subfractions (F1–F8). Subfraction F4 displayed bactericidal activity (Figure 3a). The crude extract from the BSB1 fermentation medium was not fractionated, due to the small quantity (0.32 g). However, it was subjected to chromatographic analysis.

#### 3.9.2. Fourier Transform Infrared (FT-IR)

We placed 50 µL of the EtOAc extracts between two NaCl cells, and then the solvent was evaporated. The thin film formed was measured under ambient conditions, in the midinfrared region, using the transmittance mode. Scanning conditions were as follows: a spectral range of 4000–400 cm^−1^ and a resolution of 4 cm^−1^. We recorded 64 scans and corrected them against ambient air as the background. Measurements were carried out with a NICOLET 380 FT-IR (Thermo Fisher Scientific, Verona Road, Madison, WI, USA).

#### 3.9.3. Nuclear Magnetic Resonance (NMR)

The bioactive subfraction F4 was assessed with nuclear magnetic resonance (NMR) spectrophotometry in a VARIAN AS400 MHz Mercury plus NMR spectrometer system (Varian, Inc., Palo alto, CA, USA), with 400 MHz for 1H and 101 MHz for ^13^C. The spectra were recorded in CDCl_3_ at 25 °C.

#### 3.9.4. Gas Chromatography–Mass Spectrometry (GC–MS)

The total extract from BSB1, and the active subfraction F4 from BCB11 were derivatized with *N*-methyl-*N*-(trimethylsilyl)trifluoroacetamide (MSTFA, Sigma–Aldrich, Buchs, Switzerland) before the gas chromatography–mass spectrometry (GC–MS) analysis. For this purpose, 5.0 mg of the sample was dissolved in 50 µL of MSTFA (0.27 mmol) and incubated at 60 °C for 20 min while being shaken at 450 rpm in a ThermoMixer (Eppendorf AG, Hamburg, Germany). Before injection, the silylated sample was diluted with 50 µL of EtOAc, and was placed in a chromatographic vial containing a glass insert. The analysis was carried out using a 6890 N gas chromatograph (Agilent Technologies, Palo Alto, CA, USA) equipped with a fused silica capillary column (30 m × 0.25 mm × 0.25 µm film thickness) with 5%-phenyl-dimethylpolysiloxane utilized as stationary phase (HP-5MS, J&W Agilent). The detector was an Agilent 5973 MSD mass spectrometer.

The injection volume was 1.0 µL and the carrier gas flow was He at 23.5 mL/min under splitless injection conditions. The initial oven temperature, 100 °C, was maintained for 1.0 min, and then raised to 280 °C at 10 °C/min and maintained for 10 min. Other settings included a 280 °C interface temperature, 230 °C ion source temperature, and electron impact ionization (EI) at 70 eV. The mass spectrum was analyzed, in the range of 40–600 atom mass units (amu), at a rate of 2.62 scans/s, for a run time of 29 min. MS data were processed using the Enhanced Data Analysis program. The mass spectra of the compounds encountered in the samples were matched with the National Institute of Standards and Technology (NIST 1.7 and NIST MS search 2.3) and WYLEY 275 libraries.

### 3.10. Statistical Analysis

All experiments were performed with three independent replicates (*n* = 3) and the data are expressed as means ± standard deviation. The results were analyzed via one-way ANOVA, and significant differences between the mean values were determined using Duncan’s multiple range test (*p* < 0.05). The STATGRAPHICS Centurion XV statistical package (Statgraphics Technologies, Plains, VA, USA) was applied.

## 4. Conclusions

GC–MS analysis identified metabolite profiles for two rhizospheric bacterial strains, which proved to be quite similar in terms of chemical structures and which contained metabolites with demonstrated bactericidal activity. Fermentation of the TSB culture medium, using *Enterobacter* species isolated from the rice rhizosphere, produced extractable metabolites in EtOAc with promising bactericidal activity. This indicated its potential for fighting *B. glumae*. The *Enterobacter* strains were isolated from different paddy fields, one under the influence of *B. glumae*, and they showed slightly different metabolomic profiles. This suggests that the presence of the *B. glumae* pathogen could induce differentiation in the evaluated microorganisms’ metabolic pathways.

The bioassay-guided method revealed that the main compound, 3-phenylpropanoic acid, identified in the EtOAc extracts from fermented biomass with *Enterobacter* strains, was one of the metabolites responsible for the bactericidal activity exerted against *B. glumae* (EF193638.1). Concerning the capacity of the isolated rhizobacteria as PGPR, the presence of phytohormones in the EtOAc extract, together with the demonstrated ability of these rhizobacterial strains to solubilize inorganic phosphorus, indicates that they are microorganisms that could potentially improve plant growth. Further investigation of the possible application of these bioactive secondary metabolites in rice cultivation, as well as their potential ability to control other phytopathogens, is suggested.

## Figures and Tables

**Figure 1 molecules-25-02567-f001:**

Inorganic phosphorus solubilization by rice-associated rhizobacteria, (**a**) untreated control, (**b**) inorganic phosphorus solubilization activity performed by BSB1, (**c**) inorganic phosphorus solubilization activity performed by BCB11.

**Figure 2 molecules-25-02567-f002:**
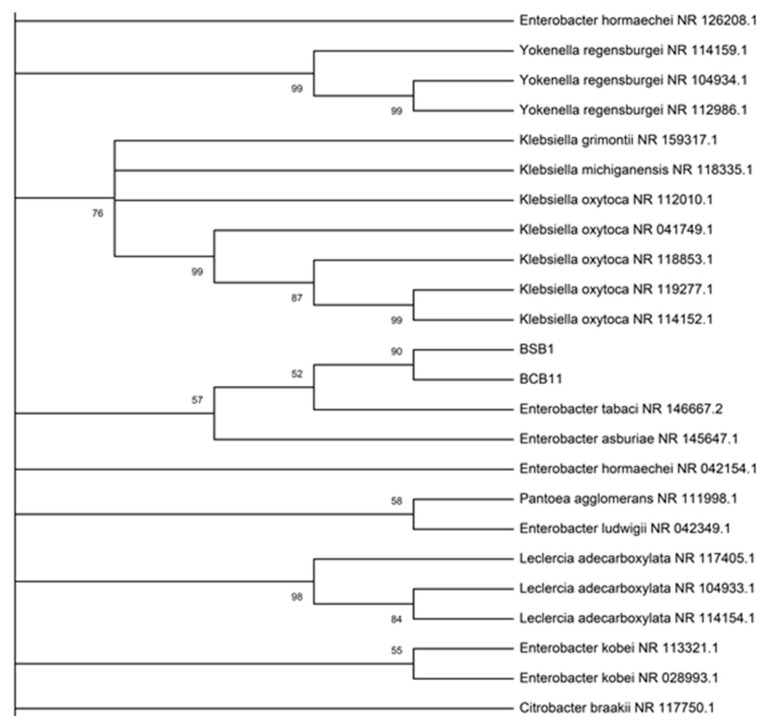
Phylogenetic tree of BSB1 (*MK715467*) and BCB11 (*MK715464*) strains, based on 16S rDNA sequence analysis, and constructed using the neighbor-joining method. The level of bootstrap values (expressed as a percentage of 1000 repetitions) is indicated at all nodes to be greater than 50%.

**Figure 3 molecules-25-02567-f003:**
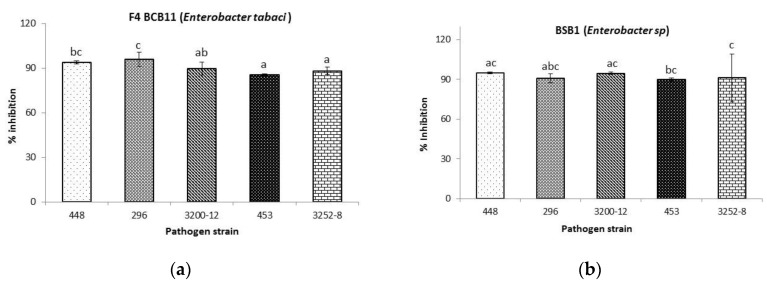
Growth inhibition of different *B. glumae* strains. (**a**) Fraction F4 BCB11 (*Enterobacter tabaci* (*E. tabaci*)) inhibitory activity. (**b**) Inhibitory activity of total EtOAc extract from *Enterobacter sp.* fermentation. Different letters indicate significant differences (*p* < 0.05), following Duncan’s multiple range test.

**Figure 4 molecules-25-02567-f004:**
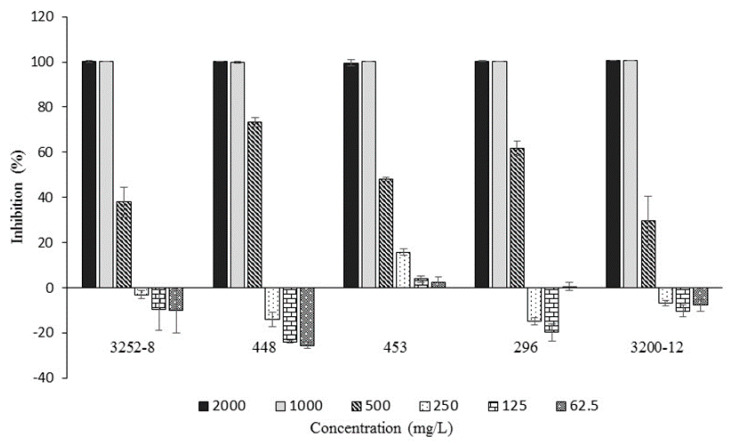
Growth inhibition of different *B. glumae* strains produced by different concentrations of 3-phenylpropanoic acid.

**Table 1 molecules-25-02567-t001:** The inhibition halozones produced by antagonistic rhizobacteria toward *Burkholderia glumae* (*B. glumae*).

Rhizobacteria	*Burkholderia glumae* Strains				
	296	448	453	3200-12	3252-8
	Halozone (mm)				
BCB11	9.1 ± 1.6 ^c^	N.D	N.D	N.D	7.6 ± 1.2 ^b^
BSB1	6.7 ± 0.8 ^b^	5.7 ± 0.9 ^b^	N.D	N.D	6.8 ± 1.2 ^b^
Control	22.1 ± 1.9 ^a^	17.3 ± 0.7 ^a^	27.3 ± 2.0	32.9 ± 3.8	21.0 ± 1.9 ^a^

N.D: Not detectable. Values followed by different letters in a single column were found to be significantly different (*p* < 0.05), following Duncan’s multiple range test. The control was a 50 mg/L solution of oxolinic acid.

**Table 2 molecules-25-02567-t002:** Seed germination and seedling development.

Bacteria	Germination %	Shoot Length (cm)	Root (cm)	Seedling Weight Dry (mg)	Shoot/Root Ratio (s/r)
	“FEDEARROZ 67”				
BCB11	76.67 ± 4.71 ^a^	6.55 ± 0.97 ^a^	4.49 ± 0.84 ^a^	10.69 ± 0.30 ^a^	1.46 ± 0.33 ^a^
BSB1	90.00 ± 8.16 ^b^	5.02 ± 0.76 ^b^	4.40 ± 0.68 ^a^	12.02 ± 0.32 ^b^	1.14 ± 0.26 ^a^
Control	70.48 ± 8.19 ^a^	6.35 ± 0.92 ^a^	4.26 ± 0.72 ^a^	11.19 ± 0.78 ^a,b^	1.49 ± 0.26 ^a^
	**“FEDEARROZ 2000”**				
BCB11	96.67 ± 4.71 ^a^	6.46 ± 0.90 ^a^	6.90 ± 1.00 ^b^	14.94 ± 2.73 ^a^	0.93 ± 0.18 ^a^
BSB1	96.67 ± 4.71 ^a^	7.42 ± 0.93 ^b^	7.71 ± 1.12 ^a^	14.69 ± 2.15 ^a^	0.96 ± 0.18 ^a^
Control	96.67 ± 4.71 ^a^	6.96 ± 1.03 ^a,b^	7.95 ± 1.19 ^a^	15.95 ± 1.70 ^a^	0.87 ± 0.19 ^a^

Values followed by different letters in a single column were found to be significantly different (*p* < 0.05), under Duncan’s multiple range test. Controls were made with 2.0 mL of a 10 mM sterile MgSO_4_ solution on filter paper.

**Table 3 molecules-25-02567-t003:** Identification of metabolites by GC–MS ^1^ in the ethyl acetate extracts of the fermentation medium of *Enterobacter* strains.

Compound	Retention Time (min)	BCB11	BSB1
**1**	3.408	Lactic Acid	Lactic Acid
**2**	5.498	Benzoic acid	Benzoic acid
**3**	6.108	Benzeneacetic acid	Benzeneacetic acid
**4**	6.270	Butanedioic acid	Butanedioic acid
**5**	6.396	-	Catechol
**6**	6.643	Uracil	Uracil
**7**	6.770	-	2,5-dihydroxy-3,6-dihydro-3,6-dimethylpyrazine
**8**	6.815	Nonanoic acid	Nonanoic acid
**9**	7.455	2,4-Dihydroxy-5-methyl-pyrimidine	-
**10**	7.630	3-phenylpropanoic acid	3-phenylpropanoic acid
**11**	7.742	beta-Alanine	beta-Alanine
**12**	7.926	-	Indole
**13**	8.017	Decanoic acid	Decanoic acid
**14**	8.650	4-Hydroxybenzyl alcohol	4-Hydroxybenzyl alcohol
**15**	8.715	*N*-(2-phenylethyl)-acetamide	*N*-(2-phenylethyl)-acetamide
**16**	8.863	*N*-Acetylphenylethylamine	-
**17**	9.114	Cinnamic acid	-
**18**	9.449	Tyrosol	Tyrosol
**19**	9.642	*N*-Phenethylpropionamide	-
**20**	10.223	4-Hydroxybenzeneacetic acid	4-Hydroxybenzeneacetic acid
**21**	11.570	Phloretic acid	Phloretic acid
**22**	11.665	-	Benzyl benzoate
**23**	12.408	Myristic acid	Myristic acid
**24**	13.024	Tryptophol	Tryptophol
**25**	13.326	-	8-Phenyloctanoic acid
**26**	13.711	3-Indolacetic acid	3-Indolacetic acid
**27**	14.346	Palmitic acid	Palmitic acid
**28**	14.730	(*Z*)-octadec-9-enenitrile	(*Z*)-octadec-9-enenitrile
**29**	14.839	3-Indolepropionic acid	-
**30**	15.228	5-Hydroxytryptophol	5-Hydroxytryptophol
**31**	15.919	Oleic Acid	Oleic Acid
**32**	16.129	Stearic acid	Stearic acid
**33**	18.216	(*Z*)-Docos-9-enenitrile	-

^1^ The compounds with carboxylic, hydroxyl, and amine groups have been identified in the GC–MS as the corresponding trimethylsilyl derivatives (TMS derivative).

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
