# Peer review of "Rice-Associated Rhizobacteria as a Source of Secondary Metabolites against *Burkholderia glumae"

_molecules, 2020, doi:10.3390/molecules25112567_

Round 1

Reviewer 1 Report

Dear authors

Many thanks for your submission to Molecules MDPI journal.

Here, my major concerns:

  1. You mentioned that the objective of the present study was to identify metabolites, which would inhibit the growth of various strains of B. glumae. However, you did not assessed any isolated metabolites for this objective. You only isolated active fractions. Please, how can you explain this major concern?
  2. To complement Table 1, please, can you provide some pictures to illustrate the inhibitory effect produced by BCB11 and BSB1 (vs controls)?
  3. It is not clear for me how did you conduct the germination and seedling growth experiments? Were they two different experiments? or, did you determine shoot length, root length, seedling weight dry and shoot/root ratio using the germination experiment? 
  4. To complement Table 2, please, can you provide some pictures to illustrate the effects produced by BCB11 and BSB1 on seed germination, shoot length, root length and seedling weight dry?
  5. Why did not you use a positive control (e.g. Azotobacter chroococcum as you mentioned) in your experiments? You discussed the obtained results in the work with those reported in reference 27 and 30. But, were similar the experimental conditions?
  6. Phylogeny in current form is not admisible. Please, I invite to the authors to prepare a robust and accurate phylogeny including a high number of strains in the analysis.
  7. How were emulsified the EtOAc extracts obtained from the promissory strains? 
  8. Please, explain in detail what is the point with the metabolic analysis. For example: you emphasize a lot the use of the 3-phenylpropanoic acid as antibacterial compound; however, you did not test this compound as pure metabolite. I don´t understand this section in the manuscript.

Reviewer 2 Report

The manuscript "Rice-associated rhizobacteria as a source of secondary metabolites against Burkholderia glumae" by Atuesta et al. characterizes the plant growth-promoting capacity of two strains that present antagonistic activity against Burkholderia glumae. In addition, the authors identify the metabolomics profiles of these strains and evaluate the potential of one of the compounds identified in the antibacterial activity against B. glumae.

Although the article presents metabolomic data of these strains with antagonistic activity against B. glumae, which is innovative and will help to better understand how and what the mechanism behind the antagonistic activity of these strains, the article has many flaws and is lacking of appropriate discussion to be accepted for publication in its current state. The manuscript needs to be thoroughly revised, there are meaningless phrases or even the meaning of several paragraphs in the introduction is poor, the results are not properly discussed and the methods need to be better described and even divided into different sections. 

Round 2

Reviewer 1 Report

Dear authors:

Many thanks for your kind responses.

Regarding responses:

Point 1: You did not modify the text in the reviewed manuscript according with my comment. I invite you to reflect about my point.

Point 6: Please, check again the phylogeny with an expert. You can present a more accurate phylogeny.

Reviewer 2 Report

The revised version of the manuscript "Rice-associated rhizobacteria as a source of secondary metabolites against Burkholderia glumae" by Atuesta et al. reveals that the authors made significant changes to the text resulting in a significant improvement in its scientific quality.

The introduction was reviewed in depth as well as the presentation and discussion of the results. The article in general has good scientific quality and addresses an interesting and innovative approach to combating Burkholderia glumae.

For these reasons, I am of the opinion that the article has undergone significant changes that put it with quality to be accepted in molecules.

There are minor flaws in the text that can be improved. For instance:

Keywords- "PGP" should be spell out

L38- Despite being in the abstract, "BPB" should be written in full the first time it is mentioned in the introduction.

L79-L88- From "Further" to " iron-chelating activity [29]" should be one single paragraph.
